# A quantum chemical approach representing a new perspective concerning agonist and antagonist drugs in the context of schizophrenia and Parkinson's disease

Ana Martínez [1,2] *, Ilich A. Ibarra [3], Rubicelia Vargas [2]

**1** Instituto de Investigaciones en Materiales, Universidad Nacional Autónoma de México, Circuito Exterior S. N., Ciudad Universitaria, CDMX, México, **2** Departamento de Química, División de Ciencias Básicas e Ingeniería, Universidad Autónoma Metropolitana-Iztapalapa, CDMX, México, **3** Laboratorio de Fisicoquímica y Reactividad de Superficies (LaFReS), Instituto de Investigaciones en Materiales, Universidad Nacional Autónoma de México, CDMX, Mexico

* martina@unam.mx

**Data Availability Statement:** All relevant data are within the manuscript and its Supporting Information files.

## Abstract

Schizophrenia and Parkinson's disease can be controlled with dopamine antagonists and agonists. In order to improve the understanding of the reaction mechanism of these drugs, in this investigation we present a quantum chemical study of 20 antagonists and 10 agonists. Electron donor acceptor capacity and global hardness are analyzed using Density Functional Theory calculations. Following this theoretical approach, we provide new insights into the intrinsic response of these chemical species. In summary, antagonists generally prove to be better electron acceptors and worse electron donors than dopamine, whereas agonists present an electron donor-acceptor capacity similar to that of dopamine. The chemical hardness is a descriptor that captures the resistance of a chemical compound to change its number of electrons. Within this model, harder molecules are less polarizable and more stable systems. Our results show that the global hardness is similar for dopamine and agonists whilst antagonists present smaller values. Following the Hard and Soft Acid and Bases principle, it is possible to conclude that dopamine and agonists are hard bases while antagonists are soft acids, and this can be related to their activity. From the electronic point of view, we have evolved a new perspective for the classification of agonist and antagonist, which may help to analyze future results of chemical interactions triggered by these drugs.

## Introduction

Schizophrenia, a type of psychosis, is associated with diverse symptoms such as avolition, catatonia, diminished emotional expression, anhedonia, disorganized speech, delusions, hallucinations and psychomotor abnormality [1–10]. The main hypothesis that could explain schizophrenia is related to the neurotransmitter dopamine. In the dopaminergic system

**Funding:** The authors received no specific funding for this work.

**Competing interests:** The authors have declared that no competing interests exist.

dopamine is synthesized in dopaminergic nerve terminals from the amino acid tyrosine. Then it is absorbed by a vesicular monoamine transporter where is stored until it is used during neurotransmission, which is regulated by the receptors. The dopamine hypothesis of schizophrenia states that the physiological mechanism evolves from an excess of dopamine activity in certain regions of the brain and little dopamine activity in other regions. Several drugs named antipsychotics have been developed to control the symptoms of schizophrenia, which primarily target this dopaminergic pathway [11–37].

The first antipsychotic to be developed in the modern *era* was chlorpromazine (*Largactil* ®), introduced in the 1950's [28–32]. Since then, dozens of drugs with this capacity have been synthesized and tested. The so-called First-Generation Antipsychotics (FGA) are effective for controlling the symptoms of schizophrenia, but they have detrimental extra-pyramidal side effects that curtail long-term treatments. Atypical antipsychotics or Second-Generation Antipsychotics (SGA) had been used since 1989, when clozapine was introduced to the US market [33–35]. The common characteristic of SGA is that they are not only effective for controlling hallucinations and delusions, but also considerably reduce extra-pyramidal side effects. Concerning the prevention of schizophrenia symptoms, SGA can be distinguished from FGA because the former manifest more than one action mechanism. Pharmacodynamic properties reflect SGA affinity for serotonin and dopamine specific receptors. Experiments show that most of the FGA and SGA are antagonists of dopamine and serotonin. They occupy the receptors but do not activate them.

The Third-Generation of Antipsychotics (TGA) includes aripiprazole and cariprazine. Unlike other antipsychotics, these molecules are not dopamine antagonists but agonists, *i.e.* they occupy and activate the receptors [17–18]. However, in the presence of high extracellular concentrations of dopamine, these drugs compete with dopamine, while also acting as antagonists with many clinical benefits. In summary, they are named "dopamine stabilizers", as they are agonists in regions with low dopamine concentrations, whereas in areas with high dopamine concentrations, these drugs act as antagonists. Nonetheless, the main action of these drugs is as agonists.

Parkinson's disease is also related to an imbalance in the level of dopamine. The dopamine precursor L-DOPA (L-3,4-dihydroxyphenylalanine) was previously successfully used for the treatment of this disorder; however long-term exposure to this drug accelerates the dopamine neurodegenerative process [38–43]. Dopamine receptor agonists such as pramipexole delay complications associated with exposure to L-DOPA and have been successfully used in therapy for Parkinson's disease [44, 45].

Drugs related to schizophrenia and Parkinson's disease bind to different dopamine receptors that belong to the family of G-Protein-Coupled Receptors (GPCR). Previous investigations have modeled dopamine receptors, providing important information about the mechanism of ligand-receptor non-covalent interactions [46–56]. Nevertheless, little is known about the intrinsic reactivity of these drugs. In spite of all reports concerning the function of antagonists and agonists, and also studies that consider the receptors, there are no quantum chemistry investigations that compare these two, or that consider different reactivity parameters. It is now well known that antipsychotics act on the receptor of dopamine to alleviate psychosis. Apparently, all approved antipsychotics present some affinity for these receptors. However, these molecules also bind to other receptors. As there is more than one dopamine GPCR participating, it is important to characterize the drugs independently of the receptors. Light bulbs and sockets provide a good analogy to explain the importance of studying these drugs. Some light bulb characteristics are independent of the sockets (for example, light bulbs can have different voltage). If we consider that GPCR are the sockets and the drugs are the light bulbs, it follows that certain characteristics of the drugs are independent of the receptors.

Following these ideas, the main goal of this work is to characterize 20 antagonists and 10 agonists of dopamine as electron donors or as electron acceptors, by using intrinsic reactivity indexes within the context of Chemical Reactivity Theory in DFT (CRDFT) [57–62]. The binding energies of these antipsychotics with dopamine receptors are already reported [46, 53] but we think it still needs to be explained why some drugs behave as agonists and others as antagonists from the quantum chemistry point of view. The intrinsic properties of these drugs may provide some useful insights. Global hardness (η) of each molecule is also a reactivity index that is also reported as a possible parameter useful for the characterization of agonist and antagonist. The conclusions obtained from these reactivity indexes offer a new perspective for the analysis of these drugs.

## Methods

Gaussian09 was used for all electronic calculations [63] Geometry optimizations without symmetry constraints were implemented at M06/6-311+G(2d,p) level of theory [64–68] while applying the continuum solvation model density (SMD) with water, in order to mimic a polar environment [69]. Harmonic analyses were calculated to verify local minima (zero imaginary frequencies). Geometries from PubChem database were employed as initial structures for the geometry optimization. Other conformers of each molecule were also optimized but the optimized ground states are those that come from PubChem [70]. Cartesian coordinates of more stable optimized molecules are included as Supporting Information. We also considered protonated states for those molecules with pKa values lower than 7.

In order to analyze electron-donor acceptor properties, vertical ionization energy (I) and vertical electron affinity (A) were obtained from single point calculations of the corresponding cationic and anionic molecules, using the optimized structure of the neutrals. The same level of theory was used for all computations.

In this investigation we analyze 20 antagonists and 10 agonists of dopamine (see Fig 1). These molecules have been chosen due to the following reasons [11]. The selected First-Generation Antipsychotics represent different families of compounds. Haloperidol, trifluperidol, benperidol, spiperone, pipamperone and droperidol are typical antipsychotic of the butyrophenone family. They exhibit high affinity dopamine D2 receptor antagonism and slow receptor dissociation kinetics. These butyrophenones have different receptor binding profiles and exhibited distinctive clinical efficacy. Haloperidol is one of the first drugs used to treat schizophrenia and it is the most commonly used. Trifluperidol is stronger than haloperidol. Benperidol and spiperone are two of the most potent antipsychotics in this family. Moreover, spiperone is used in treating drug-resistant schizophrenia. Pipamperone present a different pharmacological profile and it can be classified as a first-generation typical antipsychotic. It was considered as a forerunner of atypical antipsychotics. Chlorprothixene and clopenthixol are typical antipsychotics of the thioxanthene group. Raclopride is a selective antagonist of dopamine receptors and it can be radiolabelled and used as a tracer for *in vitro* imaging. Sulpiride belongs to the benzamide class. Within this group of molecules, we can compare ligands of the same family and also ligands from different families.

The First-Generation Antipsychotics have largely given way to Second-Generation Antipsychotics such as risperidone, clozapine, olanzapine, quetiapine and ziprasidone. Clozapine was the first antipsychotic of the second generation that was synthesized and tested. It has antipsychotic action but no Parkinson-like motor side effects and it is one of the most widely used together with risperidone. Clozapine is the most efficacious antipsychotic drug and it is used only for treatment of resistant schizophrenia due to its severe side effects. Olanzapine has a similar structure and it is also a good antipsychotic but with lower side effects. Risperidone,

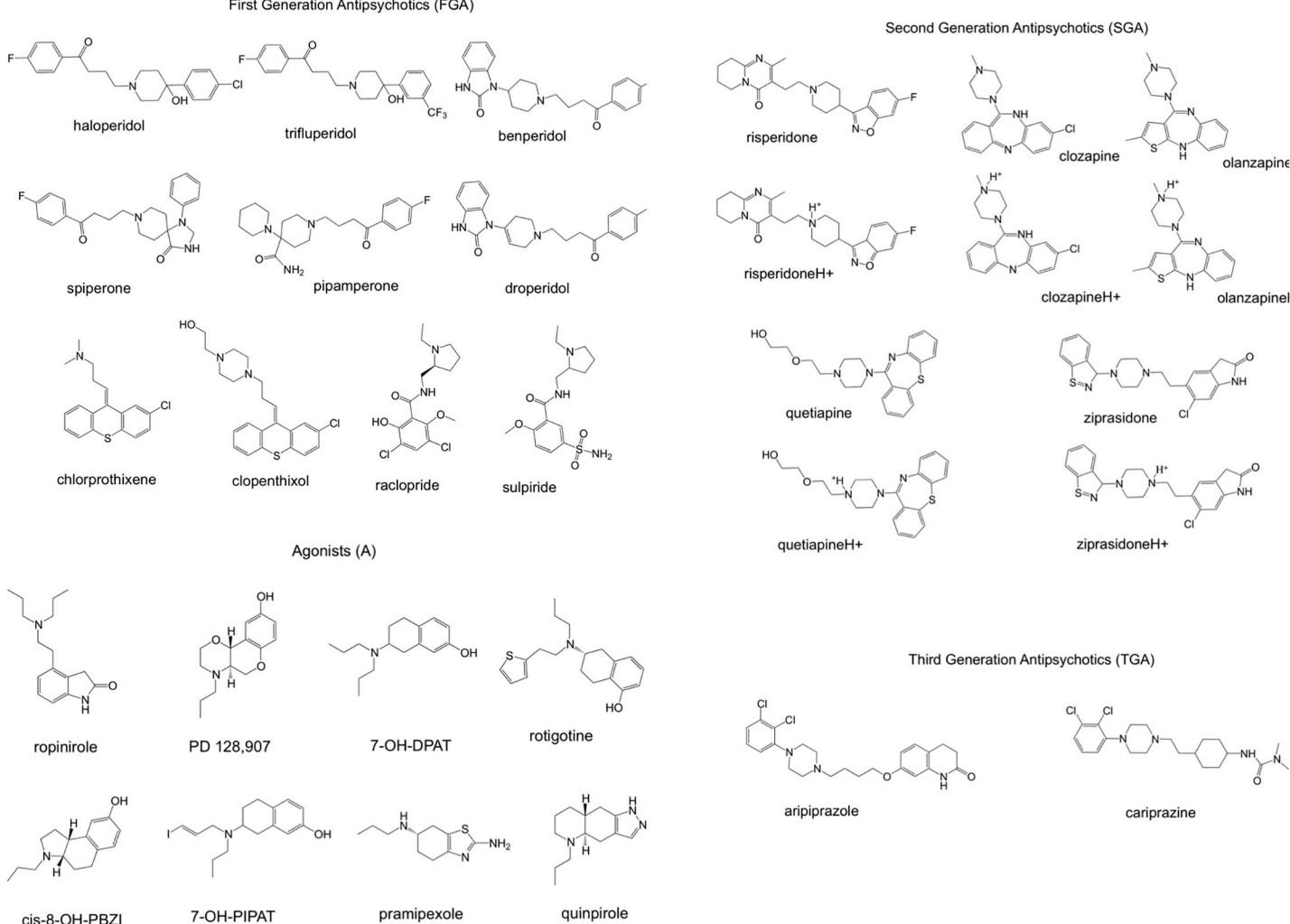

**Fig 1. Molecular formulas.** Molecules that we analyzed in this investigation are presented. To mimic physiological conditions, protonated species are included for those molecules that have pKa values lower than 7.

the second atypical antipsychotic quickly became a first-line treatment for acute and chronic schizophrenia because of its preferential side effect profile. Ziprasidone was also developed based upon clozapine and it is used since 1998. All these antipsychotics are widely used in the treatment of schizophrenia.

Aripiprazole and cariprazine are relatively new antipsychotic drugs and represents the Third-Generation of Antipsychotics. They are dopamine system stabilizers. Finally, the agonists that we investigated were selective developed to be agonists and therefore, they are very well characterized as agonists. These molecules were investigated theoretically with docking studies that allow investigating potential sites of interaction [53].

## Results and discussion

### Electron transfer process

All molecules have chemical properties that can be described in terms of response functions. These response functions refer to modifications in the electronic states of one molecule due to

the presence of other molecules. For chemical interactions that are mainly driven by electron transfer processes, these functions have been proven to qualitatively describe and explain fundamental aspects of chemical reactivity [71–75]. In this model, an electron bath constitutes the chemical environment in which chemical species are immersed.

In this investigation, we analyzed the drugs by studying the global response on the specific section of molecules when they are immersed in an idealized bath that may either donate or accept charge. The main target is to classify the drugs as either electron donors or acceptors. The hypothesis is that those molecules that are agonists to dopamine should have electron transfer properties similar to this neurotransmitter. They will therefore interact to the receptors and activate them (as dopamine does). Molecules that are antagonists of this neurotransmitter must have a different capacity to transfer charge. They should be different electron-donor acceptors of dopamine, which explains why they interact to the receptors but do not activate them.

The response functions that we used in this investigation are the electrodonating ($w^-$) and electroaccepting ($w^+$) powers, previously reported by Gázquez *et al* [61,62]. These authors defined the propensity to donate charge or $\omega^-$ as follows:

$$\omega^- = (3I + A)^2/16\,(I-A) \tag{1}$$

whereas the propensity to accept charge or $\omega^+$ is defined as

$$\omega^+ = (I + 3A)^2/16\,(I-A) \tag{2}$$

I and A are vertical ionization energy and vertical electron affinity, respectively. Lower values of $\omega^-$ imply greater capacity for donating charge. Higher values of $\omega^+$ imply greater capacity for accepting charge. In contrast to I and A, $\omega^-$ and $\omega^+$ refer to charge transfers, not necessarily from one electron. This definition is based on a simple charge transfer model expressed in terms of chemical potential and hardness. The Donor-Acceptor Map previously defined [71] is a useful graphic tool that has been used successfully in many different chemical systems [72–75]. We have plotted $\omega^-$ and $\omega^+$ (Fig 2) on this map, enabling us to classify substances as either electron donors or acceptors. Electrons are transferred from good donor systems (down to the left of the map) to good electron acceptor systems (up to the right of the map).

Fig 3 presents the DAM for antipsychotics and drugs for Parkinson's disease (antagonists and agonists) that have been used for several years. Corresponding molecular structures are included in Fig 1. We should remember that FGAs and SGAs are both antagonists, whereas TGAs are partial agonists. We included other molecules reported as dopamine agonist in the analysis [46,53]. As can be seen in Fig 3, agonists are located close to dopamine in the DAM. This means that they are more like dopamine than antagonists. *i.e.* they are good electron donors being the best quinpirole. The best electron acceptors (up to the right) are FGAs classified as butyrophenones, an important family of compounds with antipsychotic properties. In the middle of the map we found SGAs and other FGAs. They are not as good electron donors as neurotransmitter and they are also worse electron acceptors than butyrophenones. The TGAs are molecules that are partial agonists and are located close to dopamine in the DAM. It is clear that these agonists or antagonists present different capacity to either donate or accept electrons, and correspondingly, they could show different efficacy for controlling schizophrenia or Parkinson's disease.

In summary, analysis of the DAM shows that antagonists are better electron acceptors and worse electron donors than dopamine. Within this model, these drugs act as antagonist, as they do not have the same capacity to accept or donate electrons as dopamine. Conversely,

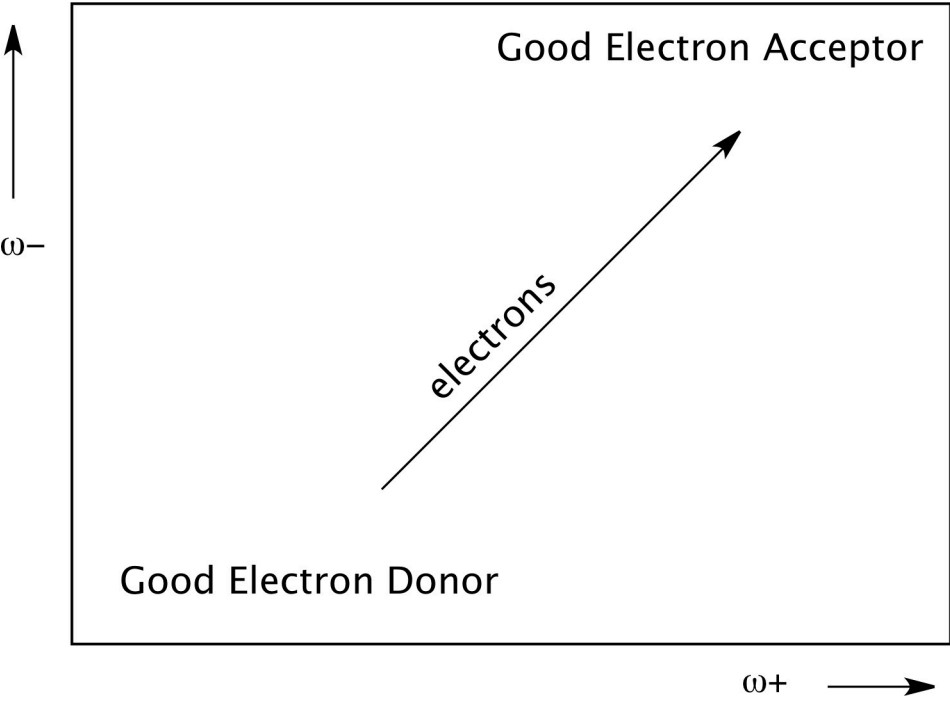

**Fig 2. DAM.** Donor-Acceptor Map.

they manifest the opposite, which may relate to the fact that they block the receptor without activating it. Instead, the agonists studied here present similar potential for donating electrons, which is possibly an indication of the activation capacity of these drugs.

Our model is based on the intrinsic reactivity indexes that can be obtained from quantum chemistry calculations, allowing us to classify these drugs with precision as either electron donors or electron acceptors. Agonists have similar electron donor capacity as dopamine. Contrastingly, antagonists are better electron acceptors and worse electron donors than dopamine. This model may also explain the partial agonist action of TGA, as the electron donor acceptor capacity of these molecules is similar to that of dopamine (they are close to this molecule in the DAM); it thus represents a better electron donor and worse electron acceptor than other antipsychotics. In summary, within this model we suggest a classification of these drugs based on the charge transfer process.

## Chemical hardness ($\eta$)

One of the main goals of quantum chemistry is to offer accessible and elegant methods capable to explain complicated chemical processes. Over the last three decades, theoretical chemists have developed concepts and principles in order to understand different reactions and interactions. Pearson introduced one interesting and qualitatively model in the early 1960´s [57–59] as an attempt to unify the concepts of organic chemical reactivity with inorganic chemistry reactions. Pearson's model is named as the Hard and Soft Acid and Base (HSBA) principle and it is an elegant theory widely used in chemistry to explain stability, reaction mechanisms and also, to describe the sensitivity and performance of explosive materials [76]. In 1983, Pearson and Parr [60] reported a quantitative definition within DFT framework of the chemical hardness ($\eta$) that is approximated as follows (I and A are vertical ionization energy and vertical

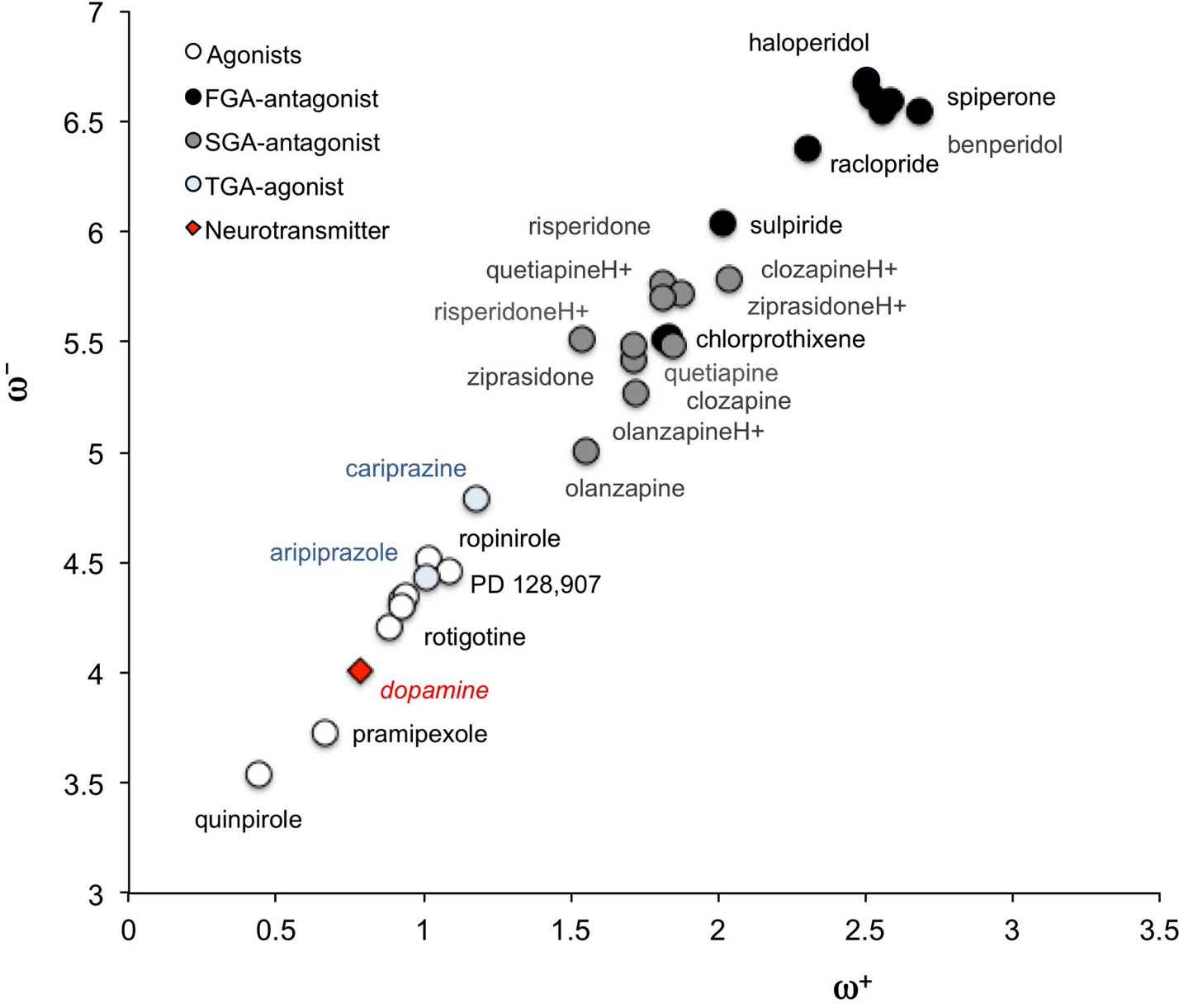

**Fig 3. DAM of molecules under study.** Donor-Acceptor Map of antagonists and agonists. Dopamine is included for comparison.

electron affinity, respectively)

$$\eta = I - A \qquad (3)$$

Hardness refers to stability and polarizability of the molecules. It captures the resistance of a chemical species to modify the number of electrons [57–60,62]. A Lewis acid is an electron acceptor and a base is an electron donor. In general, HSBA states that *soft acids react and form stronger bonds with soft bases, whereas hard acids are mainly bonded to hard bases*. In attempt to give more insights to characterize agonists and antagonists, chemical hardness was obtained for all the molecules that we are investigating. Fig 4 reports these results. FGAs (black rhombus) and SGAs (grey rhombus) show lower η than dopamine (the exception is risperidoneH⁺).

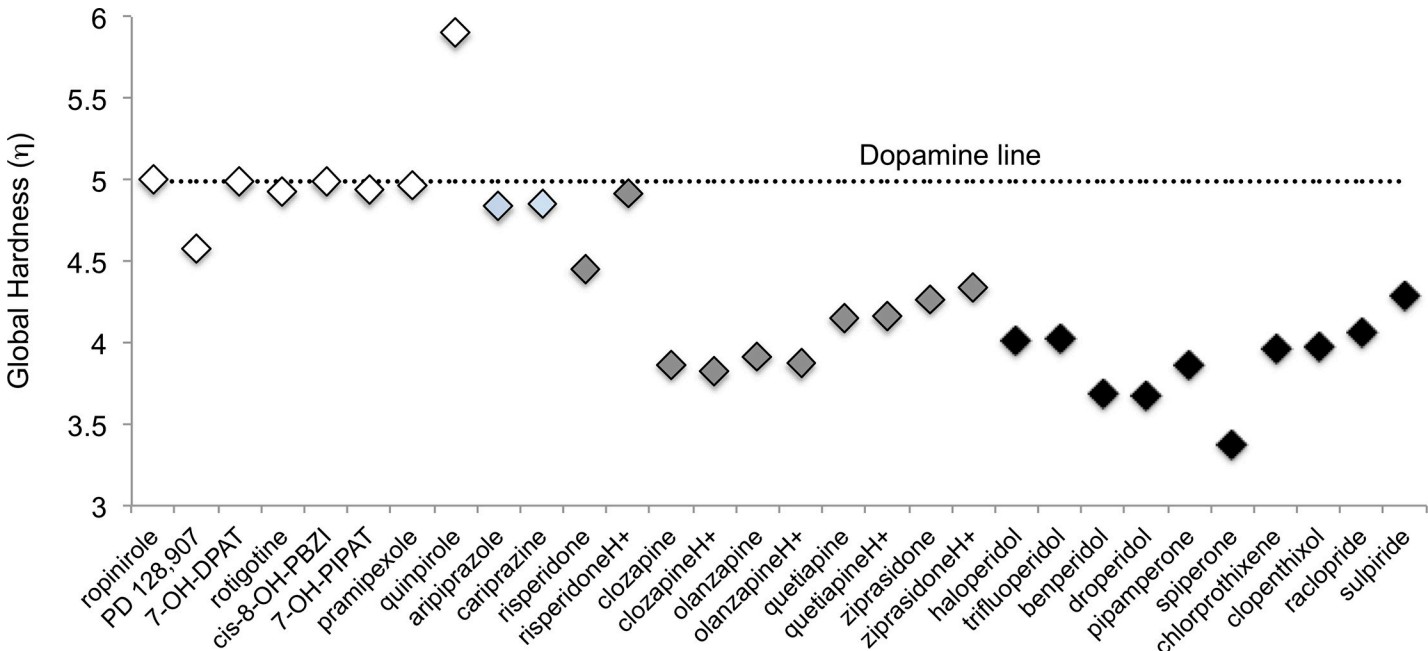

**Fig 4. Global hardness (η).** Global hardness for all the molecules studied in this investigation. FGAs (black rhombus) SGAs (grey rhombus) TGAs (blue rhombus) and agonists (white rhombus).

TGAs (blue rhombus) and must of the agonists (white rhombus) have similar values of η. According to these results, antagonists are softer and more polarizable molecules than dopamine, whereas most agonists are similar to dopamine, *i.e.* agonists and dopamine are less polarizable than antagonists that may undergo polarizable.

Analyzing together the electron-donor acceptor capacity from the previous section with the chemical hardness, it is possible to conclude that dopamine and agonists are hard molecules and also good electron donors. Antagonists are soft molecules and good electron acceptors. Following the HSBA principles, dopamine and agonist are hard bases whilst antagonists are soft acids. According to this principle, dopamine and agonists react in a similar way and this could be related to the ability to activate the receptor. On the other hand, antagonists react different since they are soft acids, and this can explain why they block the receptors and fail to activate them. Thus, within this model we suggest that hardness and the electron donor-acceptor properties may be useful to classify these drugs. The classification as hard and soft acids and bases offers a smart methodology, which helps us to categorize agonists and antagonists.

## Conclusions

The intrinsic reactivity-index using quantum chemistry calculations provide a precise classification of these drugs as hard/soft acid/bases.

Within the HSBA principles, dopamine and agonist are hard bases while antagonists are soft acids. Dopamine and agonists respond in a similar way and this could be related to the ability to activate the receptor. Antagonists are soft acids and this can explain why they block the receptors and fail to activate them.

We proposed a computational model able to classify agonist and antagonist, based on an analysis of the intrinsic reactivity of the molecules. This provides a new perspective in order to describe antagonists and agonists related with schizophrenia and Parkinson's disease. We

anticipate that this model will help to understand future experimental and theoretical results concerning to these significant systems.

## Supporting information

**S1 Data. Cartesian coordinates of the optimized structures.**
(PDF)

## Acknowledgments

This study was funded by DGAPA-PAPIIT, Consejo Nacional de Ciencia y Tecnología (CONACyT), and resources provided by the Instituto de Investigaciones en Materiales (IIM). This work was carried out using a NES supercomputer, provided by Dirección General de Cómputo y Tecnologías de Información y Comunicación (DGTIC), Universidad Nacional Autónoma de México (UNAM). We would like to thank the DGTIC of UNAM for their excellent and free supercomputing. We also thank the Laboratorio de Supercómputo y Visualización en Paralelo at the Universidad Autónoma Metropolitana- Iztapalapa for the access to its computer facilities. Authors would like to acknowledge Oralia L Jiménez, María Teresa Vázquez and Cain González for their technical support.

## Author Contributions

**Conceptualization:** Ana Martínez, Ilich A. Ibarra, Rubicelia Vargas.

**Investigation:** Ana Martínez, Rubicelia Vargas.

**Writing – original draft:** Ana Martínez, Rubicelia Vargas.

**Writing – review & editing:** Ana Martínez, Ilich A. Ibarra.

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
