## [Decision Letter · Decision Letter 0]

10 Sep 2019

PONE-D-19-20849

A quantum chemistry approach representing a new perspective concerning agonist and antagonist drugs in the context of schizophrenia and Parkinson’s disease

PLOS ONE

Dear Prof. Martinez Vazquez,

Thank you for submitting your manuscript to PLOS ONE. After careful consideration, we feel that it has merit but does not fully meet PLOS ONE’s publication criteria as it currently stands. Therefore, we invite you to submit a revised version of the manuscript that addresses the points raised during the review process.

Two reviewers indicate that revisions are required.  Both of them have raised the issue of bringing in the properties of the receptors in a clear way. Please respond to all of the points raised by both reviewers by revision and/or rebuttal. I will send your resubmission to the same reviewers for reassessment.

We would appreciate receiving your revised manuscript by Oct 25 2019 11:59PM. To enhance the reproducibility of your results, we recommend that if applicable you deposit your laboratory protocols in protocols.io, where a protocol can be assigned its own identifier (DOI) such that it can be cited independently in the future. For instructions see: http://journals.plos.org/plosone/s/submission-guidelines#loc-laboratory-protocols

We look forward to receiving your revised manuscript.

Kind regards,

Dennis Salahub

Academic Editor

PLOS ONE

Journal Requirements:

1. Please amend either the title on the online submission form (via Edit Submission) or the title in the manuscript so that they are identical.

Reviewers' comments:

Reviewer's Responses to Questions

**Comments to the Author**

1. Is the manuscript technically sound, and do the data support the conclusions?

Reviewer #1: Yes

Reviewer #2: Yes

2. Has the statistical analysis been performed appropriately and rigorously? 

Reviewer #1: Yes

Reviewer #2: No

3. Have the authors made all data underlying the findings in their manuscript fully available?

Reviewer #1: Yes

Reviewer #2: Yes

4. Is the manuscript presented in an intelligible fashion and written in standard English?

Reviewer #1: Yes

Reviewer #2: Yes

5. Review Comments to the Author

Reviewer #1: I congratulate the authors for the work, however, in the present work the authors using Electron donor acceptor capacity and global hardness are analyzed using Density Functional Theory calculations. Following this theoretical approach, the authors provide new insights into the intrinsic response of these chemical species. Thus, the method presented might mean a new contribution to the different criteria, however, before acceptance for publication I would recommend important changes to be taken into account in the manuscript. After careful evaluation of your article, there are some numbers of comments to be addressed follows:

1. The author should describe the dopaminergic pathway details in the introduction part.

2. In the ligand-receptor interaction the authors, should describe about the ligand-receptor non-covalent interactions. Further, the drug is bonded to the receptor?

3. The electron bath is unclear in the receptors (GPCR proteins). The authors should describe the electron transfer bath and provide the image.

4. The authors claimed, the dopamine electron transfer properties are similar to this neurotransmitter but the authors did not describe about the similarities.

5. The authors should provide the clear image about the “Agonists have similar electron donor capacity and therefore they may also transfer electrons to the receptors and activate them. Contrastingly, antagonists would accept electrons from the receptors, impeding activation”.

Reviewer #2: The authors present their thesis clearly. The article is well written in English. Authors analyses physico-chemistry properties of a set of molecules composed of antipsychotic drugs and agonist drug to the dopamine receptor. Properties such as acceptor of donor of electrons are computed by DFT method. The method and the atomic base-is set (M06/6-311+G(2d,p)) are well established for this kind of calculation.

My concern is about the complexity of receptors and the list of molecules.

The authors said that they are not talking about receptor. Receptors are not receptor-like in fig 4, but are receptors and family of receptors. Each molecule binds on several receptors, which are identified or even sometimes not identified. Analysis of sequences of different receptors reveals that the active site of this GPCR family is kite different. The activity of the molecule if the binding part of the mechanism is well-known this some crystallographic structure, the mechanism of the biologic response is unknown.

In conclusion of this remark, an analysis of site in term of residues should give a validation if site is fill with residues acceptor of electrons, an electron donor molecule should have a good binding.

The list of molecules studied is concise in the article. First, second, and third generations of antipsychotic drugs and agonists are more numerous than in the list of molecules. A full list is necessary to validate the assumption of agonist or antagonist drugs is correlated this the acceptor or donor of electrons. The case is more complicated than a molecule has nitrogen atom or halogen it is antagonism, and a molecule has a hydroxy function; it is an agonist.

6. PLOS authors have the option to publish the peer review history of their article (what does this mean?). If published, this will include your full peer review and any attached files.

Reviewer #1: No

Reviewer #2: No

---

## [Author Response · Author response to Decision Letter 0]

19 Sep 2019

PONE-D-19-20849

A quantum chemistry approach representing a new perspective concerning agonist and antagonist drugs in the context of schizophrenia and Parkinson’s disease

Professor Dennis R. Salahub

Academic Editor

PLOS ONE

Dear Prof. Salahub,

We thank the reviewers for their priceless evaluations and their very useful comments. Since both reviewers have raised the issue of bringing in the properties of the receptors, we would like to start explaining why we do not include the properties of the receptors.

In this investigation we provide a classification of thirty antipsychotics in terms of their electron donor-acceptor properties that give information about the intrinsic reactivity of these molecules. These characteristics of the drugs are independent of the receptors, and allow us to have a clear and straightforward classification of the antipsychotics, which, otherwise, would not be possible to achieve. Currently, a way to classify the antipsychotics is based on In Vivo Rotational Experiments with injured rats. In one of these reports, “a lesion on rats was performed on the left side of the medial forebrain bundle in the brain” and the conclusion is that “the rotations produced upon agonist Challenger were clockwise”. These experiments are extremely impressive with conclusions that are based on the behavior of four rats that have been injured. We believe that with our quantum chemical calculations, it is possible to have another classification and both classifications can complement each other.

The study of the receptors is a matter of a different investigation and in fact, there are previous studies that predict the binding sites and report binding energies of these antipsychotics with dopamine receptors (see for example references 46 and 53 of the manuscript). In spite of all the studies concerning the drugs and the receptors reported until now, there are not investigations regarding the intrinsic characteristics of these drugs. We strongly believe that it is still needed to be explained why some drugs behave as agonists and others as antagonists from the quantum chemistry point of view, and the intrinsic characteristics that we report here could give some solid ideas.

The intrinsic characteristics of these drugs allow us to classify them and also give us some ideas about possible reaction mechanisms, but it is clear that, to speak about the reaction mechanisms it is important to consider all the reactants, in this case, drugs and receptors. In this sense, the paper in the original version was confusing since we included a possible reaction mechanism and, as both reviewers established, it is not possible to analyze a reaction mechanism with only one of the reactants. In that case, the study of the GPCR proteins is also necessary. What we can do is a clear classification. For this reason, we removed, from the original version, all the information concerning the reaction mechanism, including Figure 4 and 6. Specifically, the following paragraphs were deleted in this new version:

We assume that the receptors (GPCR proteins) are represented by the electron bath and, in this context; they are able to either donate charge to the drugs or accept charge from them.

It is known that these drugs interact with GPCR through ligand-receptor non-covalent interactions but what happens once the drug is bonded to the receptor? Our premise is that once the drug interacts with the receptor, a charge-transfer process initiates. Dopamine binds to the receptors and activates them, and some of this activation may be the result of electron transfer; if this is the case, dopamine will donate electrons to the receptors.

… and therefore the may also transfer electrons to the receptors and activate them 

… would accept electrons form the receptors, impeding activation.

The model presented in here gives prominence to the charge transfer process in the action mechanism: drugs interact with the receptors and a charge transfer occurs. Dopamine activates the receptors by donating electrons to GPCRs. Agonists also transfer electrons to the receptors and activated them, as they have similar electron donor capacity. Conversely, the antagonists accept electrons from the receptors, so activation does not proceed. 

Figure 4. Schematic representation of the action mechanisms proposed in this report. Dopamine and agonist are good electron donors (represented by a red hexagon) and therefore, transfer electrons to the receptor. Antagonists (represented as a blue hexagon) are good electron acceptors and may accept electrons from the receptor.

Figure 6. Schematic representation of the action mechanism proposed in this report. Dopamine and agonist are hard molecules and good electron donors (represented by a red hexagon) and therefore they transfer electrons to the receptor without polarization. Antagonists (represented as a blue deformed hexagon) are soft molecules and good electron acceptors; they are polarizable and may accept electrons from the receptor.

Finally, in here we present and explain other modifications that were carried out on the manuscript. We hope to have been able to solve and properly answered the reviewers' concerns.

With my best regards

Prof. Ana Martínez

Reviewer #1: 

1. The author should describe the dopaminergic pathway details in the introduction part.

Authors’ reply

Following this suggestion, the first paragraph of the introduction was modified as follows:

Schizophrenia, a type of psychosis, is associated with diverse symptoms such as avolition, catatonia, diminished emotional expression, anhedonia, disorganized speech, delusions, hallucinations and psychomotor abnormality [1-10]. The main hypothesis that could explain schizophrenia is related to the neurotransmitter dopamine. In the dopaminergic system dopamine is synthesized in dopaminergic nerve terminals from the amino acid tyrosine. Then it is absorbed by a vesicular monoamine transporter where is stored until it is used during neurotransmission, which is regulated by the receptors. The dopamine hypothesis of schizophrenia states that the physiological mechanism evolves from an excess of dopamine activity in certain regions of the brain and little dopamine activity in other regions. Several drugs named antipsychotics have been developed to control the symptoms of schizophrenia, which primarily target this dopaminergic pathway [11-37].

Reviewer #1: 

2. In the ligand-receptor interaction the authors, should describe about the ligand-receptor non-covalent interactions. Further, the drug is bonded to the receptor?

Authors’ reply

We thank the reviewer for this comment that is very interesting. However, with the results reported here it is not possible to analyze the ligand-receptor interaction since the receptor is not included in this investigation. We are focused on the analysis of the drugs. 

Reviewer #1: 

3. The electron bath is unclear in the receptors (GPCR proteins). The authors should describe the electron transfer bath and provide the image.

Authors’ reply

As we explain before in this letter, we are not including the GPCR proteins in this study. 

Reviewer #1: 

4. The authors claimed, the dopamine electron transfer properties are similar to this neurotransmitter but the authors did not describe about the similarities.

Authors’ reply

We apologize for the lack of clarity in the explanation. We added the following paragraph

As can be seen in Figure 2, agonists are located close to dopamine in the DAM. This means that they are more like dopamine than antagonists. i.e. they are good electron donors being the best quinpirole.

Reviewer #1: 

5. The authors should provide the clear image about the “Agonists have similar electron donor capacity and therefore they may also transfer electrons to the receptors and activate them. Contrastingly, antagonists would accept electrons from the receptors, impeding activation”.

Authors’ reply

As we pointed out before, in this new version we removed all the information concerning the reaction mechanism.

Reviewer #2: 

The authors said that they are not talking about receptor. Receptors are not receptor-like in fig 4, but are receptors and family of receptors. Each molecule binds on several receptors, which are identified or even sometimes not identified. Analysis of sequences of different receptors reveals that the active site of this GPCR family is kite different. The activity of the molecule if the binding part of the mechanism is well-known this some crystallographic structure, the mechanism of the biologic response is unknown. In conclusion of this remark, an analysis of site in term of residues should give a validation if site is fill with residues acceptor of electrons, an electron donor molecule should have a good binding.

Authors’ reply

As we already explained previously, the main objective of this investigation is to classify the antipsychotics by using intrinsic properties. It is a good idea to include residues in order to analyze the binding scheme, but this is a matter of a different investigation. The classification presented here could be very useful for future investigations and also in the clinic of these patients.

Reviewer #2: 

The list of molecules studied is concise in the article. First, second, and third generations of antipsychotic drugs and agonists are more numerous than in the list of molecules. A full list is necessary to validate the assumption of agonist or antagonist drugs is correlated this the acceptor or donor of electrons. The case is more complicated than a molecule has nitrogen atom or halogen it is antagonism, and a molecule has a hydroxy function; it is an agonist.

Authors’ reply

We agree with the reviewer. There are hundreds of antipsychotics, but the drugs presented here are the ones that have been used for several years and they are very well characterized. Other drugs are not completely characterized, and others bind to several receptors, or are agonists/antagonist of other neurotransmitters. For some of these thirty antipsychotics there are theoretical studies concerning the binding site to the dopamine receptor. At this point we do not generalize the properties for agonists/antagonists but we classify these 30 molecules according to the electron/donor-acceptor properties.

---

## [Decision Letter · Decision Letter 1]

14 Oct 2019

PONE-D-19-20849R1

A quantum chemistry approach representing a new perspective concerning agonist and antagonist drugs in the context of schizophrenia and Parkinson’s disease

PLOS ONE

Dear Prof. Martinez Vazquez,

Thank you for submitting your manuscript to PLOS ONE. After careful consideration, we feel that it has merit but does not fully meet PLOS ONE’s publication criteria as it currently stands. Therefore, we invite you to submit a revised version of the manuscript that addresses the points raised during the review process.

Reviewer 2 requests that justification for the choice of molecules should be given.  Please give this careful consideration and revise accordingly.

We would appreciate receiving your revised manuscript by Nov 28 2019 11:59PM. To enhance the reproducibility of your results, we recommend that if applicable you deposit your laboratory protocols in protocols.io, where a protocol can be assigned its own identifier (DOI) such that it can be cited independently in the future. For instructions see: http://journals.plos.org/plosone/s/submission-guidelines#loc-laboratory-protocols

We look forward to receiving your revised manuscript.

Kind regards,

Dennis Salahub

Academic Editor

PLOS ONE

Reviewers' comments:

Reviewer's Responses to Questions

**Comments to the Author**

1. If the authors have adequately addressed your comments raised in a previous round of review and you feel that this manuscript is now acceptable for publication, you may indicate that here to bypass the “Comments to the Author” section, enter your conflict of interest statement in the “Confidential to Editor” section, and submit your "Accept" recommendation.

Reviewer #1: All comments have been addressed

Reviewer #2: All comments have been addressed

2. Is the manuscript technically sound, and do the data support the conclusions?

Reviewer #1: Yes

Reviewer #2: Yes

3. Has the statistical analysis been performed appropriately and rigorously? 

Reviewer #1: Yes

Reviewer #2: Yes

4. Have the authors made all data underlying the findings in their manuscript fully available?

Reviewer #1: Yes

Reviewer #2: Yes

5. Is the manuscript presented in an intelligible fashion and written in standard English?

Reviewer #1: Yes

Reviewer #2: Yes

6. Review Comments to the Author

Reviewer #1: All the points have been taken care of by the authors. In my opinion the manuscript is acceptable as is

Reviewer #2: The authors have an answer to previous questions. If we could agree, taking accounts the structure of receptors is hard to include in the study at this stage, the fact that one of the main actors of the interactions is missing.

On the second remark, concerning the option to take a set of well-characterized molecules, vs all molecules with molecules not well characterize is acceptable.

Nevertheless, a description of the set (30 molecules) and why each molecule is chosen is important. The point here is too validated the composition of the set and at the end the final model. The idea is too reject than it is just an ad-hoc selection of molecule which supports the clustering model.

Nevertheless, the study, which focalizes on a set of well-described ligands could be interesting for researchers in the field if the set is validated.

7. PLOS authors have the option to publish the peer review history of their article (what does this mean?). If published, this will include your full peer review and any attached files.

Reviewer #1: No

Reviewer #2: No

---

## [Author Response · Author response to Decision Letter 1]

17 Oct 2019

PONE-D-19-20849

A quantum chemistry approach representing a new perspective concerning agonist and antagonist drugs in the context of schizophrenia and Parkinson’s disease

Professor Dennis R. Salahub

Academic Editor

PLOS ONE

Dear Prof. Salahub,

We thank the useful comments of both reviewers. Reviewer 2 requests the justification for the choice of molecules. We revised the manuscript accordingly, and we added the following paragraphs at the end of the Methods section.

In this investigation we analyze 20 antagonists and 10 agonists of dopamine (see Figure 1). These molecules have been selected due to the following reasons [11]. The selected First Generation Antipsychotics represent different families of compounds. Haloperidol, trifluperidol, benperidol, spiperone, pipamperone and droperidol are typical antipsychotic of the butyrophenone family. They exhibit high affinity dopamine D2 receptor antagonism and slow receptor dissociation kinetics. These butyrophenones have different receptor binding profiles and exhibited distinctive clinical efficacy. Haloperidol is one of the first drugs used to treat schizophrenia and it is the most commonly used. Trifluperidol is stronger than haloperidol. Benperidol and spiperone are two of the most potent antipsychotics in this family. Moreover, spiperone is used in treating drug-resistant schizophrenia. Pipamperone present a different pharmacological profile and it can be classified as a first-generation typical antipsychotic. It was considered as a forerunner of atypical antipsychotics. Chlorprothixene and clopenthixol are typical antipsychotics of the thioxanthene group. Raclopride is a selective antagonist of dopamine receptors and it can be radiolabelled and used as a tracer for in vitro imaging. Sulpiride belongs to the benzamide class. Within this group of molecules, we can compare ligands of the same family and also ligands from different families. 

The First Generation Antipsychotics have largely given way to Second Generation Antipsychotics such as risperidone, clozapine, olanzapine, quetiapine and ziprasidone. Clozapine was the first antipsychotic of the second generation that was synthesized and tested. It has antipsychotic action but no Parkinson-like motor side effects and it is one of the most widely used together with risperidone. Clozapine is the most efficacious antipsychotic drug and it is used only for treatment of resistant schizophrenia due to its severe side effects. Olanzapine has a similar structure and it is also a good antipsychotic but with lower side effects. Risperidone, the second atypical antipsychotic quickly became a first-line treatment for acute and chronic schizophrenia because of its preferential side effect profile. Ziprasidone was also developed based upon clozapine and it is used since 1998. All these antipsychotics are widely used in the treatment of schizophrenia.

Aripiprazole and cariprazine are relatively new antipsychotic drugs and represents the Third Generation of Antipsychotics. They are dopamine system stabilizers. Finally, the agonists that we investigated were selective developed to be agonists and therefore, they are very well characterized as agonists. These molecules were investigated theoretically with docking studies that allow investigating potential sites of interaction [53]. 

We hope to have been able to solve and properly answered the reviewer's concerns.

Finally, we kindly would like to request if possible, a discount on the fees of this open access. Unfortunately, we do not have any funding (grants) that can cover these fees and therefore, we should pay this contribution with our own money. 

With my best regards

Prof. Ana Martínez

---

## [Editor Report · Decision Letter 2]

21 Oct 2019

A quantum chemistry approach representing a new perspective concerning agonist and antagonist drugs in the context of schizophrenia and Parkinson’s disease

PONE-D-19-20849R2

Dear Dr. Martinez Vazquez,

We are pleased to inform you that your manuscript has been judged scientifically suitable for publication and will be formally accepted for publication once it complies with all outstanding technical requirements.

With kind regards,

Dennis Salahub

Academic Editor

PLOS ONE
---

## [Editor Report · Acceptance letter]

28 Oct 2019

PONE-D-19-20849R2 

A quantum chemical approach representing a new perspective concerning agonist and antagonist drugs in the context of schizophrenia and Parkinson’s disease 

Dear Dr. Martinez Vazquez:

I am pleased to inform you that your manuscript has been deemed suitable for publication in PLOS ONE. Congratulations! Your manuscript is now with our production department. 

With kind regards,

on behalf of

Dr. Dennis Salahub 

Academic Editor

PLOS ONE